# RAC1B Regulation of *TGFB1* Reveals an Unexpected Role of Autocrine TGFβ1 in the Suppression of Cell Motility

**DOI:** 10.3390/cancers12123570

**Published:** 2020-11-29

**Authors:** Hendrik Ungefroren, Hannah Otterbein, Ulrich F. Wellner, Tobias Keck, Hendrik Lehnert, Jens-Uwe Marquardt

**Affiliations:** 1First Department of Medicine, University Hospital Schleswig-Holstein, Campus Lübeck, D-23538 Lübeck, Germany; hannah.otterbein@student.uni-luebeck.de (H.O.); Jens.Marquardt@uksh.de (J.-U.M.); 2Clinic for General Surgery, Visceral, Thoracic, Transplantation and Pediatric Surgery, University Hospital Schleswig-Holstein, Campus Kiel, D-24105 Kiel, Germany; 3Clinic for Surgery, University Hospital Schleswig-Holstein, Campus Lübeck, D-23538 Lübeck, Germany; ulrich.wellner@uksh.de (U.F.W.); tobias.keck@uksh.de (T.K.); 4University of Salzburg, A-5020 Salzburg, Austria; hendrik.lehnert@sbg.ac.at

**Keywords:** RAC1B, transforming growth factor β, pancreatic cancer, breast cancer, cell migration, autocrine regulation

## Abstract

**Simple Summary:**

Transforming growth factor (TGF)β secreted by cancer cells and acting in an autostimulatory manner has been implicated in epithelial-mesenchymal transition and cellular invasion. However, the upstream inducers/downstream effectors, or the effects on cell migration, of endogenously produced TGFβ1 remain poorly characterized. Here, we studied whether autocrine TGFβ is regulated by the small GTPase, RAC1B, and how it impacts cell motility. We found that RAC1B induced an increase in the levels of TGFβ1 mRNA and the secreted bioactive protein in pancreatic carcinoma Panc1 and breast cancer MDA-MB-231 cells. Antibody-mediated neutralization of secreted TGFβ or inhibition of its synthesis increased cell migration and the activities of promigratory genes, but downregulated those of antimigratory genes, i.e., *SMAD3*. Restoration of SMAD3 protein expression in these cells was able to prevent the TGFβ1 inhibition-induced rise in migratory activity, suggesting that a RAC1B-autocrine TGFβ1-SMAD3 pathway can suppress mesenchymal transdifferentiation and cell motility.

**Abstract:**

Autocrine transforming growth factor (TGF)β has been implicated in epithelial-mesenchymal transition (EMT) and invasion of several cancers including pancreatic ductal adenocarcinoma (PDAC) as well as triple-negative breast cancer (TNBC). However, the precise mechanism and the upstream inducers or downstream effectors of endogenous *TGFB1* remain poorly characterized. In both cancer types, the small GTPase RAC1B inhibits cell motility induced by recombinant human TGFβ1 via downregulation of the TGFβ type I receptor, ALK5, but whether RAC1B also impacts autocrine TGFβ signaling has not yet been studied. Intriguingly, RNA interference-mediated knockdown (RNAi-KD) or CRISPR/Cas-mediated knockout of RAC1B in TGFβ1-secreting PDAC-derived Panc1 cells resulted in a dramatic decrease in secreted bioactive TGFβ1 in the culture supernatants and *TGFB1* mRNA expression, while the reverse was true for TNBC-derived MDA-MB-231 cells ectopically expressing RAC1B. Surprisingly, the antibody-mediated neutralization of secreted bioactive TGFβ or RNAi-KD of the endogenous *TGFB1* gene, was associated with increased rather than decreased migratory activities of Panc1 and MDA-MB-231 cells, upregulation of the promigratory genes *SNAI1, SNAI2* and *RAC1*, and downregulation of the invasion suppressor genes *CDH1* (encoding E-cadherin) and *SMAD3*. Intriguingly, ectopic re-expression of *SMAD3* was able to rescue Panc1 and MDA-MB-231 cells from the *TGFB1* KD-induced rise in migratory activity. Together, these data suggest that RAC1B favors synthesis and secretion of autocrine TGFβ1 which in a SMAD3-dependent manner blocks EMT-associated gene expression and cell motility.

## 1. Introduction

During tumor pathogenesis the change in cell phenotype from epithelial to mesenchymal is induced by contextual signals that epithelial cells receive from the tumor microenvironment (TME) and involves a genetic program referred to as epithelial-mesenchymal transition (EMT). In primary mammary epithelial cells it was shown that the induction of the EMT program and maintenance of mesenchymal and stem cell states is enabled by paracrine and autocrine signals [1]. One of the most potent inducers of the EMT program and promoter of a mesenchymal phenotype is TGFβ signaling. TGFβ1 can stimulate its own expression [2,3] and that of its receptors [4], eventually resulting in increased sensitivity to autocrine or localized TGFβ signaling, when compared to normal epithelial cells [5]. In fact, many highly invasive carcinoma cells, i.e., the pancreatic ductal adenocarcinoma cell line, Panc1, and triple-negative breast cancer cell line, MDA-MB-231, secrete TGFβs, primarily TGFβ1, and display autocrine stimulation [2,6,7] which is believed to promote their growth, survival, invasion and eventually metastasis. However, tumor cell-specific inhibition of TGFβ signaling at either the receptor or ligand level can also enhance metastasis as shown in mouse models of metastatic breast cancer [5,8,9].

In the past couple of years it has become apparent that extensive changes in alternative splicing also play a profound role in shaping the changes in cell behavior that characterize the EMT [10]. Tumor cells benefit greatly from this flexible regulatory process since many specific isoforms have been identified as promoting and inhibiting tumor growth or invasion and metastasis in different cancer types [11]. A prototype example is *RAC1*, a gene that gives rise to two splice isoforms, RAC1 and RAC1B, belonging to the Rho family of small GTPases. RAC1 is a known driver of oncogenic transformation, tumor progression and chemoresistance of various solid cancers, in part through promoting non-Smad-dependent TGFβ signaling [12]. Likewise, RAC1B has been demonstrated to drive growth of colorectal, pancreatic, lung and thyroid carcinomas via its ability to promote cell cycle progression, apoptosis resistance, or escape from oncogene-induced senescence (reviewed in [13]. From this it follows that in those instances where cellular senescence promotes tumor initiation/progression, EMT, or metastasis [14], RAC1B may act as a tumor suppressor. However, in contrast to RAC1, the role of RAC1B in EMT and migration/invasion is less clear and partially controversial [13].

Recently, we demonstrated in pancreatic epithelial cells that RAC1B promoted the expression of *CDH1* (encoding E-cadherin, ECAD) and other epithelial genes, while inhibiting the expression of mesenchymal genes and EMT [15,16]. Mechanistically, RAC1B-dependent protection from mesenchymal conversion and acquisition of a motile phenotype was due to suppression of tumor-promoting MEK-ERK2 signaling [15,16] and interference with TGFβ1 signaling via downregulation of the TGFβ type I receptor ALK5 [4] and induction of the inhibitory Smad, SMAD7 [17]. We also observed that RAC1B upregulated SMAD3 which in its non-activated form exhibited an anti-migratory effect in pancreatic cancer cells [18] presumably due to its ability to promote the expression of ECAD, via transcriptional induction of miR-200 [19], or biglycan (BGN), a pericellular proteoglycan and potent TGFβ inhibitor [18]. Based on these findings, we have recently postulated a tumor suppressor function for RAC1B. 

Given the strong regulation of ALK5 by RAC1B, we addressed the question if this *RAC1* isoform also impacts expression of *TGFB1* and/or secretion of TGFβ1. Based on our contention that RAC1B functions as a tumor suppressor, while autocrine TGFβ1 is considered a tumor promoter, we originally hypothesized that if RAC1B indeed targets TGFβ1 this interaction will be inhibitory. Prompted by the surprising observation of RAC1B promoting TGFβ1 secretion we set out to study in more detail how endogenous TGFβ1 impacts cell motility in highly invasive tumor cells. In the end, we revealed a hitherto unappreciated role of autocrine TGFβ1 in the control of cell motility that is not only compatible with the proposed role of RAC1B as a tumor suppressor but even provides strong evidence in favor of it.

## 2. Results

### 2.1. RAC1B Promotes Expression and Secretion of TGFβ1

Previous work has shown that RAC1B negatively controls random/spontaneous cell migration (chemokinesis) in benign and malignant pancreatic and breast epithelial cells [4,16,17,18,20]. To clarify if TGFβ1 secretion was affected by the loss of RAC1B, we depleted Panc1 cells of RAC1B by either CRISPR/Cas9-mediated knockout (Panc1^RAC1BKO^) or subjected Panc1 and MDA-MB-231 cells to RNA interference-mediated knockdown (RNAi-KD) resulting in Panc1^RAC1BKD^ and MDA-MB-231^RAC1BKD^ cells, respectively. A TGFβ1 enzyme-linked immunosorbent assay (ELISA) was then performed to determine the relative amount of biologically active TGFβ1 released by these RAC1B-depleted cells into the media during a 24 h period. It should be noted that the bioactive TGFβ1 in cell culture supernatant may not be completely reflective of all secreted TGFβ1 as a fraction of it may bind to the extracellular matrix if sufficient fibronectin is assembled. Interestingly, RAC1B depletion resulted in strongly reduced levels of bioactive TGFβ1 in the culture supernatants of both Panc1^RAC1BKO^ and Panc1^RAC1BKD^ cells (Figure 1A). Conversely, the levels of bioactive TGFβ1 in culture supernatants of MDA-MB-231 cells stably transfected with a HA-tagged version of RAC1B (MDA-MB-231^HA-RAC1B^) were higher than in those from empty vector controls (Figure 1B) and, surprisingly, chemokinetic activities of HA-RAC1B-expressing MDA-MB-231 cells were lower than those of control cells as determined by real-time cell migration assay (Appendix A). 

To reveal whether stimulation of autocrine TGFβ1 secretion by RAC1B involves transcriptional regulation of the TGFB1 gene, the genetically engineered cells from above were screened by quantitative real-time RT-PCR (qPCR) for alterations in steady-state mRNA expression of TGFβ1 and the related TGFβ2 and 3. Interestingly, in Panc1^RAC1BKO^ cells TGFβ1 the mRNA level was reduced to 39.1 ± 19.6% (*p* < 0.002, *n* = 5) of that in empty lentivirus (LV) control cells (Figure 1C). Likewise, transcript levels of the other two TGFβs were also lower in Panc1^RAC1BKO^ cells (Figure 1C). In contrast, the abundance of mRNA for the β_A_ subunit of activins A/AB (encoded by INHBA) was increased in Panc1^RAC1BKD^ cells (Figure 1C) showing that RAC1B was not generally stimulatory for all members of the TGFβ superfamily of ligands. A decrease in TGFβ1 mRNA following RAC1B depletion was also seen in MDA-MB-231^RAC1BKD^ cells (59.8 ± 21.4% of control, *p* < 0.05, *n* = 3, Figure 1D), albeit to a lesser extent which may be due to the low expression of RAC1B in these cells. Together, these results suggest that in pancreatic and breast carcinoma cells synthesis and secretion of bioactive TGFβ1 is positively controlled by RAC1B. Since we were particularly interested in RAC1B regulation of TGFβ signaling and carcinoma cells predominantly express TGFβ1 [8], we focussed on TGFB1 in all further experiments.

### 2.2. Effect of Neutralizing Autocrine TGFβ on Cell Migration and Gene Expression

In light of our previous findings, the observation of RAC1B promoting TGFβ gene expression and secretion was somehow counterintuitive as TGFβ1 is considered a protumorigenic factor and autocrine TGFβ1 has been shown to drive malignancy. Based on our hypothesis that RAC1B is a tumor suppressor gene, we nevertheless speculated that induction of endogenous TGFB1 may serve anti- rather than protumorigenic functions. This idea was further supported by the finding of lower migration activity scores in MDA-MB-231^HA-RAC1B^ vs. control cells despite the presence of higher levels of secreted TGFβ1 in the culture medium. We therefore decided to inactivate the secreted TGFβ(1) in the culture medium by antibody-mediated neutralization, thereby preventing binding to its cognate receptors and potential autocrine stimulation. Following incubation of MDA-MB-231 cells with a pan-specific anti-TGFβ1/2/3 antibody, but not vehicle or IgG1 isotype control, their chemokinetic activity was found to be significantly enhanced (Figure 2A). Of note, the changes in migration behavior following incubation with the anti-TGFβ antibody was associated with an increase in the expression of the promigratory genes SNAI1 and RAC1 and a decrease in expression of the antimigratory SMAD3 (Figure 2B). Similar data were obtained with Panc1 cells albeit the migratory response to anti-TGFβ antibody treatment occurred with slower kinetics (Appendix A). To achieve robust effects, a ten-fold higher concentration (50 µg/mL) of this antibody was required which likely reflects higher secretion of TGFβ1 by Panc1 compared to MDA-MB-231 cells [6,7,21]. 

### 2.3. Endogenous TGFB1 Mimics the Suppressive Effect of RAC1B on Cell Motility

To confirm the above results and to explore the possibility that autocrine TGFβ1 mediates the inhibitory effect of RAC1B on cell motility, we performed RNAi-KD of the endogenous TGFβ1 gene in MDA-MB-231 or Panc1 cells followed by real-time measurements of cell migration. 

Intriguingly, silencing of *TGFB1* resulted in strong upregulation of basal chemokinetic activity in both cell types (Figure 3A). We went on to compare the effects on knocking down *TGFB1* with those of knocking down *RAC1B*, or those of a combined KD, in Panc1 cells and found that TGFβ1 was more potent in suppressing migration than RAC1B (Figure 3B, blue curve, tracing C vs. green curve, tracing B). Although the double-KD of *TGFB1* and *RAC1B* weakly enhanced the cells’ migratory activity over that of the *TGFB1* KD alone, the differences were neither additive nor synergistic (Figure 3B, magenta curve/tracing D vs. blue curve/tracing C). Interestingly, upon counting the transfected cells prior to subjecting them to migration assays, we consistently noted lower numbers in *RAC1B* siRNA and *TGFB1* siRNA-treated cells (Appendix A). Again, the combined KD of both proteins did not further reduce cell numbers when compared to those of the single KDs (Appendix A).

We have previously shown that Panc1 cells with stable ectopic expression of HA-RAC1B exhibited reduced basal migration [24]. Interestingly, forced RAC1B expression was associated with higher expression of *TGFB1* under basal culture conditions (Appendix A). As predicted from our hypothesis that endogenous TGFβ1 is EMT-protective, transfection of Panc1^HA-RAC1B^ cells with a TGFβ1 siRNA was capable of derepressing the chemokinetic activity of these cells to levels that reach those of the vector control cells (Figure 3C, magenta curve, tracing D vs. red curve, tracing A). These data are consistent with the results from the antibody-mediated neutralizing approach of soluble TGFβ and, in addition, reveal that endogenous production of TGFβ1 accounts for the inhibitory effect on cell migration.

### 2.4. Endogenous TGFB1 Promotes CDH1 and SMAD3, and Suppresses SNAI1 Expression 

To further corroborate the potential anti-invasive function of autocrine TGFβ1, we studied the functional consequences of inhibiting the expression of endogenous *TGFB1* or *RAC1B* on genes involved in EMT and cell motility. Much to our surprise, RNAi-KD of *TGFB1* in Panc1 or MDA-MB-231 cells—verified by qPCR and immunoblot analysis (Appendix A) and ELISA (Figure 1C, right-hand graph)—was associated with a decrease in *CDH1* mRNA expression which was more pronounced in MDA-MB-231 than in Panc1 cells (Figure 4A, upper two graphs). Again, we monitored expression of *SNAI2* in Panc1 and *SNAI1* in MDA-MB-231 cells and—in agreement with the data from the antibody-mediated inhibition of secreted TGFβ (Figure 2C and Appendix A)—found both to be significantly enhanced upon KD of either *TGFB1* or *RAC1B* (Figure 4A, lower two graphs). Interestingly, the degree of inhibition of *CDH1*, or of derepression of *SNAI2* or *SNAI1*, by a double-KD of *TGFB1* and *RAC1B* had no additive or synergistic effect compared to single-KD of each gene (Figure 4A). This finding matches well with the corresponding migration data of Panc1 cells (see Figure 3B). Positive regulation of ECAD by TGFβ1 was confirmed in Panc1^TGFB1KD^ cells by immunoblotting (Figure 4B) but was not feasible in MDA-MB-231 cells due to very low ECAD protein in these cells.

Next, we explored in Panc1^TGFB1KD^ cells by qPCR analysis if *TGFB1* alters the expression of still other genes involved in either TGFβ signaling (*TGFBR1*, *SMAD7, SMAD3*) or maintenance of a differentiated epithelial phenotype (*GRLH2*, *OVOL2*). While no significant effect was noted for *TGFBR1* (encoding ALK5) (Appendix A) or SMAD7 (Appendix A), the abundance of the *GRHL2* and *OVOL2* mRNAs was reduced (Appendix A). Intriguingly, another Smad protein, SMAD3, was downregulated upon TGFβ1 depletion at both the mRNA (Figure 4C, left-hand graph) and protein (Figure 4D) level. Conversely, RNAi-KD of SMAD3 failed to alter the levels of TGFβ1 mRNA (Figure 4C, right-hand graph) confirming that SMAD3 is induced downstream of TGFβ1 but not vice versa.

Finally, we asked if SMAD3 can mimic the stimulatory effect of endogenous TGFβ1 (or RAC1B) on ECAD abundance. To this end, RNAi-mediated silencing of *SMAD3* also resulted in decreased ECAD levels (Figure 4E). The data show that lowering endogenous expression of *TGFB1* mimics the effects of RAC1B silencing on some genes, i.e., *CDH1*, *SNAI1*, and *SMAD3* but not on others, i.e., *TGFBR1* and *SMAD7*. 

### 2.5. Ectopic Expression of SMAD3 Rescues Cells from the TGFB1 KD-Induced Increase in Cell Migration

Above, we have shown that RNAi-KD of TGFB1 or antibody-mediated neutralization of endogenous TGFβ(1) decreased the expression of SMAD3 protein, while RNAi-KD of either TGFβ1 or SMAD3 decreased abundance of ECAD. Moreover, SMAD3 inhibits cell migration independent of its C-terminally phosphorylated (=activated) form [18]. We therefore pursued the idea that SMAD3 may mediate the antimigratory effect of endogenous TGFβ1 and hence, ectopic expression of SMAD3 should rescue the cells from a *TGFB1* KD-induced increase in migratory activity. To this end, ectopic expression of SMAD3 partially rescued MDA-MB-231 and Panc1 cells from the increase in cell migration (Figure 5). Of note, treatment of Panc1^TGFB1KD^ cells with the ALK5 kinase inhibitor, SB431542 (1 µM) did not reduce their migratory activity [23]. From these data, we conclude that a RAC1B-autocrine TGFβ1-SMAD3 axis maintains ECAD expression and inhibits cell motility.

## 3. Discussion

In a series of studies we have shown that in order to efficiently inhibit EMT and cell motility RAC1B targets multiple TGFβ response genes with different functions and cellular locations such as transcription factors (*SNAI1*, *SNAI2*), intracellular signal transducers (*SMAD3*, *SMAD7*), surface receptors (*TGFBR1*, *F2RL1*) and secreted factors (*SERPINE1*, *BGN*). In a previous study, we had already shown that KD of RAC1B led to upregulation of the related RAC1 at both the mRNA and protein level [16]. This prompted us to analyse if RAC1B also impacts the bona fide ligands of the TGFβ receptors, TGFβ1, 2 and 3. Based on data from other cellular systems on the tumor promoting role of autocrine TGFβ signaling in cancer cells [5,7,25,26], we expected to find an inhibitory effect of RAC1B on TGFβ gene expression and/or secretion. However, much to our surprise, we learned that RAC1B enhanced rather than decreased both endogenous mRNA production of all three TGFβ isoforms and secretion of bioactive TGFβ1 protein. Given the close association of RAC1B expression with a well-differentiated, epithelial phenotype in a series of PDAC [15,18] and breast cancer [27] derived cell lines, we speculated that autocrine TGFβ1 might have an hitherto unappreciated role in maintaining the epithelial phenotype or preventing mesenchymal differentiation and cell motility.

We observed that the use of a (pan-specific) neutralizing antibody to the TGFβs, added to the culture medium of MDA-MB-231 or Panc1 cells, enhanced their migratory activities. This suggests that the endogenously produced TGFβ needs to be secreted in a soluble form in order to execute its antimigratory effect. These quite unexpected results were confirmed by a genetic approach which showed that RNAi-KD of *TGFB1* mimicked the negative effect of RAC1B on cell motility in both Panc1 and MDA-MB-231 cells and was able to relieve the permanently suppressed migratory activity of Panc1 cells with stable ectopic expression of RAC1B. Furthermore, the combined KD of *TGFB1* and *RAC1B* on cell migration did neither produce an additive nor a synergistic effect, suggesting that RAC1B and TGFβ1 act through the same pathway to suppress cell migration.

An interesting issue that is currently under investigation in our laboratory relates to the question of whether RAC1B targets autocrine TGFβ1 in a direct or indirect fashion. Given the small increase in total levels of the parental isoform, RAC1 in response to RAC1B KD [16], along with still preliminary data in Panc1 cells showing that selective KD of RAC1 *enhanced* autocrine TGFβ1 secretion, it is conceivable that repression of RAC1 is involved in promoting autocrine TGFβ1 by RAC1B (Figure 6). If so, this raises the possibility that RAC1B can act as an endogenous inhibitor of RAC1 in autocrine TGFβ1 production. However, we also noted that ectopic expression of a constitutively active RAC1 mutant in Panc1 cells failed to alter (downregulate) SMAD3 protein levels (Appendix A). This suggests that in the absence of concomitant RAC1B KD, RAC1 alone is unable to mediate the inductive effect of RAC1B on SMAD3 [18]. The role of RAC1 in RAC1B-driven autocrine TGFβ production/secretion, SMAD3 expression and cell migration definitely needs further attention.

Given the close association of a cells’ migratory potential with changes in morphology, gene expression and epigenetic marks collectively referred to as EMT, we addressed the question of whether autocrine TGFβ1 would affect established markers of this process. Intriguingly, RNAi-KD of *TGFB1* (this study, Figure 4) was associated with a decrease in expression of the epithelial marker and invasion suppressor ECAD, and an increase in expression of the master EMT transcription factors SLUG and SNAIL. In the same series of experiments, we observed that endogenous TGFβ1, like RAC1B [18], promoted the expression of SMAD3 (Figure 4). Hence, RAC1B, autocrine TGFβ1 and SMAD3 form a pathway—with *TGFB1* being positioned downstream of *RAC1B* but upstream of *SMAD3* (Figure 6)—that maintains high ECAD expression in cells. Promotion of ECAD expression by ectopically expressed SMAD3 was demonstrated before in a gastric cancer cell line [19]. These authors concluded that endogenous TGFβ signaling was not involved in the upregulation of EMT markers by SMAD3 in this Smad3-reconstituted cell line since both treatment with recombinant TGFβ or the ALK5 kinase inhibitor SB431542 failed to affect the expression of ECAD (or miR-200). However, based on the results of this study it is conceivable that autocrine TGFβ is an upstream activator of SMAD3-miR-200-driven ECAD expression (Figure 6). Moreover, the RAC1B-TGFβ1-SMAD3 pathway may extend to epithelial genes other than *CDH1* as we observed positive regulation by *TGFB1* of *GRHL2* and *OVOL2*. Both genes encode transcription factors that control the expression of *CDH1* [28], regulate epithelial morphogenesis, differentiation and plasticity [29], and suppress EMT-driven tumor progression in part via a reciprocal feedback loop with ZEB1 [30,31]. Like RAC1B, they also block ERK1/2 MAPK activation and inhibit TGFβ-induced EMT [32] by interfering with TGFβ signaling at multiple levels, including inhibition of *DPC4*/*SMAD4* and induction of *SMAD7*.

RAC1B utilizes various mechanisms to interfere with productive TGFβ signaling such as upregulation of SMAD7 and subsequent SMAD7-mediated downregulation of ALK5 [4,17]. We would like to speculate that in late-stage tumors characterized by a TGFβ-rich environment these inhibitory interactions may operate to decrease the cells’ sensitivity to paracrine stromal cell-derived TGFβ (Figure 6). Moreover, both RAC1B [16] and SMAD3 [18] jointly induce the pericellular proteoglycan BGN that sequesters TGFβ in the TME to inhibit its biological activity (Figure 6). It remains to be tested if autocrine TGFβ is also involved in this process. The results of this study are compatible with another variation of the theme, inaccessibility of ALK5 for stromal cell-derived TGFβ by autocrine TGFβ-directed desensitization of the pathway (Figure 6). Indeed, exposure of MDA-MB-231 and other cell types to TGFβ produced a transient response that attenuated over time, resulting in desensitized cells that were refractory to further acute stimulation. This loss of signaling competence depended on ligand binding, but not on receptor activity, and was restored only after the ligand had been depleted. The TGFβ binding triggered the rapid depletion of signaling-competent receptors from the cell surface through internalization combined with delayed membrane recruitment [33]. Based on their experimental findings and using a biochemical response, C-terminal phosphorylation of Smad2, as a readout of receptor activity, the authors derived at a computational model of TGFβ signal transduction from the membrane to the nucleus. Intriguingly, this model predicts that autocrine signaling, such as that associated with tumorigenesis, severely compromises the TGFβ response due to desensitization [33]. Therefore, a tumor that receives a constant supply of TGFβ, either because of autocrine signaling or through production of TGFβ by stromal cells [34], maintains only a low level of signaling. This would allow expression of a subset of TGFβ target genes while concomitantly silencing those genes that require a strong acute signal. This data is in accordance with observations that TGFβ’s effects can be concentration-dependent or that different cellular responses to TGFβ may require distinct signaling thresholds [35]. In our experimental setting the endogenously produced and secreted TGFβ is believed to be insufficient to activate the invasive program. Only after ligand consumption or neutralization, or inhibition of ligand synthesis, cells are resensitized to further (acute) ligand stimulation, i.e., by the exogenous TGFβ contained in the serum supplement of the growth medium. In support of this scenario, we have preliminary data to indicate that blockage of autocrine TGFβ1 production by RNAi-KD of *TGFB1* dramatically enhances the cells’ migratory response to subsequent stimulation with recombinant human TGFβ1 [23]. The results of our study are, thus, in good agreement with the prediction of the suggested model, namely that autocrine TGFβ can be protective against the action of exogenous/stromal cell-derived TGFβ and, in addition, extend its validity to a crucial cellular response, cell migration/invasion (Figure 6). Similar findings demonstrating the migration-inhibitory effects of autocrine TGFβ1 have been reported earlier in human corneal fibroblasts [35], Langerhans cells in skin [36], and rat and human vascular smooth muscle cells [37]. Taken together, all three RAC1B-driven mechanisms discussed above and summarized in Figure 6 may combine to assure effective prevention of ALK5 activation and SMAD signaling by stromal cell-derived paracrine TGFβ, and thus protection from mesenchymal transdifferentiation and acquisition of a motile phenotype.

Both RAC1B [38] and autocrine TGFβ [7] have been implicated in promoting cell growth/proliferation and KD of either protein reduced cell numbers to a similar extent (Appendix A). Although rescue experiments still need to be performed, it is tempting to speculate that autocrine TGFβ also mediates the inhibitory effect of RAC1B on TGFβ1-induced growth arrest observed earlier in Panc1^RAC1BKO^ cells [4]. Moreover, since the prosurvival effect of RAC1B is mediated by NFκB [38,39] and NFκB transcriptional activity has been shown to increase autocrine TGFβ1 [40], it is conceivable that NFκB links RAC1B-mediated growth and survival to escape from TGFβ1-induced growth arrest or apoptosis via autocrine TGFβ.

Tumor cells secrete growth factors in an autocrine manner to protect themselves from spontaneous differentiation and to support their self-renewal [1]. However, the reverse scenario, secretion of factors to protect themselves from spontaneous (or induced) dedifferentiation has been less well studied. As outlined above, RAC1B, autocrine TGFβ1 and SMAD3 act together to maintain epithelial gene expression and protect cells from mesenchymal conversion by exogenous (stromal cell-derived) TGFβ. As discussed earlier, SMAD3 induced ECAD in a gastric cancer cell line following transcriptional induction of the miR-200b/a promoter and subsequent inhibition of ZEB1 expression [19]. Hence, the miR-200/ZEB1 autoregulatory loop represents a central knot on which pro- and anti-EMT pathways converge [41,42], and in breast cancer cells an autocrine TGFβ/ZEB/miR-200 signaling network regulates plasticity between epithelial and mesenchymal states [43]. From an evolutionary perspective it is also intriguing that only slight modifications in the proteins of this newly discovered pathway in either amino acid composition, phosphorylation status, or level of expression fundamentally changes their function from pro- to anti-EMT and from agonists to antagonists of TGFβ signaling. In case of *RAC1* this is achieved by alternative splicing with inclusion of only 17 amino acids leading to the generation of RAC1B rather than RAC1. In contrast to C-terminally phosphorylated SMAD3, the non-phosphorylated form promotes ECAD expression and inhibits migration, while low-level autocrine TGFβ can antagonize the action of high-level stromal cell-derived TGFβ.

On the basis of encouraging preclinical work showing therapeutic benefit of targeting the TGFβ signaling axis, over 40 early-phase clinical oncology trials are now ongoing, using various TGFβ pathway antagonists either as single agents, or in combination with other therapeutics, including immune checkpoint inhibitors (https://clinicaltrials.gov) [44,45]. However, preclinical data were derived from a relatively small number of well-studied models that fail to capture the heterogeneity of the human disease. Other serious issues relate to the heavy reliance on immunodeficient mice, and the failure to use metastatic burden as the clinically relevant endpoint. To overcome some of these issues, Yang and coworkers recently tested the effect of anti-TGFβ-neutralizing antibodies on the metastatic endpoint in a panel of 12 immunocompetent allograft models of metastatic breast cancer [8]. Of note, they observed heterogeneous responses with 5 of 12 models showing an immune-dependent suppression of metastasis, while another 3 models responded to anti-TGFβ therapy with an undesirable stimulation of metastasis, which was immune-independent and targeted the tumor cell compartment. Intriguingly, KD of SMAD3, but not SMAD2, in one model with stimulated metastasis (MVT1) increased metastasis, suggesting that SMAD3 has direct anti-metastatic effects on the tumor cell in this model [8]. The mouse model data therefore suggest that anti-TGFβ antibodies can promote metastasis by interfering with metastasis-suppressive effects of TGFβ on the tumor parenchyma and that tumor-suppressive responses to TGFβ may be retained and dominant in some instances of advanced metastatic breast cancer, and possibly other cancers [8]. When viewed together with the data of our study, this raises the possibility that some patients would be at risk to have their disease course accelerated on anti-TGFβ therapy. Larger scale preclinical studies and good predictive biomarkers will be crucial to the safe and effective use of TGFβ pathway antagonists in clinical oncology trials.

## 4. Materials and Methods

### 4.1. Cells and Generation of Stable Clones Overexpressing RAC1B

The PDAC-derived cell lines Panc1 and the TNBC-derived cell line MDA-MB-231 were maintained in RPMI 1640 supplemented with FBS, 1% penicillin-streptomycin-glutamine (Life Technologies, Darmstadt, Germany) and 1% sodium pyruvate (Merck Millipore, Darmstadt, Germany). Both cell lines belong to the quasi-mesenchymal/basal-like subtype [46], which has been shown to utilize transcriptionally-dominated programs to lose their epithelial phenotype during EMT [46]. For this reason, changes in ECAD expression in response to (autocrine) TGFβ inhibition could be faithfully monitored by qPCR. Panc1^RAC1BKO^ cells were generated by CRISPR/Cas technology through deletion of exon 3b of *RAC1*, which had been verified by qPCR [4]. Panc1 cells stably expressing a HA-tagged version of RAC1B in the pCGN vector (Panc1^HA-RAC1B^), or empty vector as control, were selected with hygromycin B and described in detail earlier [24]. MDA-MB-231 cells stably transfected with HA-RAC1B were generated by the same limited dilution procedure and individual clones, or a mixed pool of individual clones for vector controls, selected with 500 μg/mL hygromycin B (Sigma Aldrich, Merck KGaA, Darmstadt, Germany).

### 4.2. Reagents

For immunological detection we used the following primary antibodies: anti-Smad3, #ab40854, Abcam (Cambridge, UK), monoclonal anti-TGFβ/1/2/3 neutralizing antibody (clone 1D11) or the isotype-matched IgG1 monoclonal antibody (clone 11711, all from R&D Systems, Wiesbaden, Germany), anti-TGF β1 (3C11) antibody, sc-130348, anti-TGFβ RI (V22) antibody, #sc-398, and anti-HSP90, #sc-13119, all from Santa Cruz Biotechnology (Heidelberg, Germany), anti-Rac1b, #09-271, Merck Millipore), anti-Rac1, cat.#610650, BD Transduction Laboratories (Heidelberg, Germany), anti-GAPDH (14C10), #2118, Cell Signaling Technology (Frankfurt am Main, Germany), and anti-HA, #1583816, Roche (Mannheim, Germany). HRP-linked anti-rabbit, #7074, and anti-mouse, #7076, secondary antibodies were from Cell Signaling Technology. Recombinant human TGF-β1, #300-023, was purchased from ReliaTech (Wolfenbüttel, Germany) and SB431542 from Merck. 

### 4.3. Transient Transfection of siRNA and Expression Vectors

For transient transfection, cells were seeded on day 1, cells into Nunclon^TM^ Delta Surface plates (Nunc, Roskilde, Denmark), and transfected twice, on days 2 and 3, with either 25 or 50 nM of prevalidated siRNAs specific for RAC1B [24], SMAD3 (both from Dharmacon, Lafayette, CO, USA), or the respective scrambled controls. The TGFB1 siRNA was provided by Qiagen (cat.# 1027416, a mixture of four different pre-evaluated siRNAs), and the SMAD3 siRNA by Dharmacon (Lafayette, CO, USA). Additional validation of these siRNAs was performed for RAC1B [24] and SMAD3 [18]. An expression vector for full length human TGFβ1 (cat.# SC119746) was purchased from OriGene Technologies Inc. (Rockville, MD, USA), while a MYC-tagged version of constitutively active RAC1 (Q61L mutation) in the pRK5 vector was a kind gift of G.M. Bokoch (La Jolla, CA, USA). SiRNAs or plasmids were mixed with Lipofectamine 2000 (Life Technologies) according to the manusfacturers recommendations and transfected in cells serum-free for 4 h. Following transfection with siRNAs and another 24-h incubation in normal growth medium, cells were trypsinized, diluted in trypan blue solution and viable (non-blueish) cells counted with a Neubauer chamber. 

### 4.4. QPCR Analysis

Total RNA was extracted from Panc1 cells with PeqGold RNAPure (Peqlab, Erlangen, Germany). For each sample, 2.5 μg RNA were reverse transcribed for 1 h at 37 °C, using 200 U M-MLV Reverse Transcriptase and 2.5 μM random hexamers (Life Technologies) in a total volume of 20 μL. Target gene mRNA expression was quantified by qPCR on an I-Cycler (BioRad, Munich, Germany) with Maxima SYBR Green Mastermix (Thermo Fisher Scientific, Waltham, MA, USA) and data were normalized to the expression of GAPDH. PCR primer sequences are given in Appendix A.

### 4.5. Immunoblotting

Immunoblot analysis was performed as described previously [6,16]. In brief, cells were washed once with ice-cold PBS to remove serum proteins and lysed with 1× PhosphoSafe lysis buffer (Merck Millipore). Following clearance of the lysates by centrifugation, their total protein content was determined with the DC Protein Assay (BioRad). Proteins were fractionated by polyacrylamide gel electrophoresis on mini-PROTEAN TGX any-kD precast gels (BioRad) and blotted to PVDF membranes. Membranes were blocked with nonfat dry milk or bovine serum albumin and incubated with primary antibodies overnight at 4 °C. After washing and incubation with HRP-linked secondary antibodies, chemoluminescent detection of proteins was done on a ChemiDoc XRS+ System with Image Lab Software (BioRad) with Amersham ECL Prime Detection Reagent (GE Healthcare, Munich, Germany). The signals for the proteins of interest were normalized to those for the housekeeping gene GAPDH.

### 4.6. Enzyme-Linked Immunosorbant Assay for TGFβ1

Twenty-four h after the second transfection with either ctrl siRNA, RAC1B or TGFβ1 siRNA transfected Panc1 cells, or Panc1^RAC1BKO^ or MDA-MB-231^HA-RAC1B^ cells, received fresh medium containing 0.5% FBS and culture supernatants were allowed to be conditioned for another 24 h. Aliquots from the culture supernatants were cleared by centrifugation, acidified, and after dilution subjected to a TGFβ1-specific ELISA (Human/Mouse TGF beta1 ELISA Ready-SET-Go!, eBioscience/Affymetrix Inc., San Diego, CA, USA) according to the manufacturer’s instructions. Data for bioactive TGFβ1 were normalised to cell numbers. The detection limit for total TGFβ1 was 25 pg/mL.

### 4.7. Migration Assays

We employed the xCELLigence^®^ DP system (ACEA Biosciences, San Diego, distributed by OLS, Bremen, Germany) to measure chemokinesis of Panc1 and MDA-MB-231 cells. Briefly, CIM plates-16 were prepared as detailed in the instruction manual and previous publications [16,24]. The underside of the upper chambers of the CIM plate-16 was coated with 30 μL of collagens I and IV in a 1:1 (*v*/*v*) mixture to facilitate adherence of the cells and enhance signal intensities. After filling the wells of the lower chambers with 170 µL of normal growth medium containing 1% FBS, the lower and upper chambers of the CIM plate-16 were assembled, the wells of the upper chamber supplied with 50 µL of the same medium and the whole device equilibrated in the incubator for 1 h. The upper chamber of each well was then loaded with 100 µL medium with 1% FBS containing 50,000–100,000 cells. In those experiments involving addition of anti-TGFβ1/2/3 antibody, a lower number of cells was loaded in only 50 µL of the same medium (MDA-MB-231: 10,000 cells + 5 μg/mL antibody, Panc1: 5000 cells + 50 μg/mL antibody) to assure excess of antibody over secreted TGFβ. Data were recorded at 15 or 30 min intervals, analysed with RTCA software (ACEA, version 1.2) and plotted as the dimension-less cell index (CI) reflecting relative migratory activity.

### 4.8. Statistical Analysis

Statistical significance was calculated using the Wilcoxon-test or the two-tailed, unpaired Student’s *t* test. Results were considered significant at *p* < 0.05 (*). Higher levels of significance were *p* < 0.01 (**) and *p* < 0.001 (***). 

## 5. Conclusions

Given the current concept of autocrine TGFβ as a driver of tumor progression, our observation of RAC1B promoting the expression and secretion of TGFβ1 initially appeared ad odds with its proposed role as a tumor suppressor. However, during the course of this study we realized that in mesenchymal-type cancer cells from pancreas and breast RAC1B-induced autocrine TGFβ can stimulate ECAD expression and block cell motility through a pathway that involves SMAD3 and presumably SMAD3-dependent induction of miR-200. This novel pathway may be anti-oncogenic in two ways, by desensitizing the tumor cells towards the actions of stromal cell-derived paracrine TGFβ and by promoting their epithelial phenotype via induction of SMAD3. These modes of action join in a couple of other mechanisms employed by RAC1B to potently interfere with TGFβ signaling and provides additional evidence for RAC1B being a genuine tumor suppressor. Since tumor-suppressive responses to TGFβ are retained in some advanced metastatic tumors, safe deployment of TGFβ antagonists in the clinic will require good predictive biomarkers. 

## Figures and Tables

**Figure 1 cancers-12-03570-f001:**
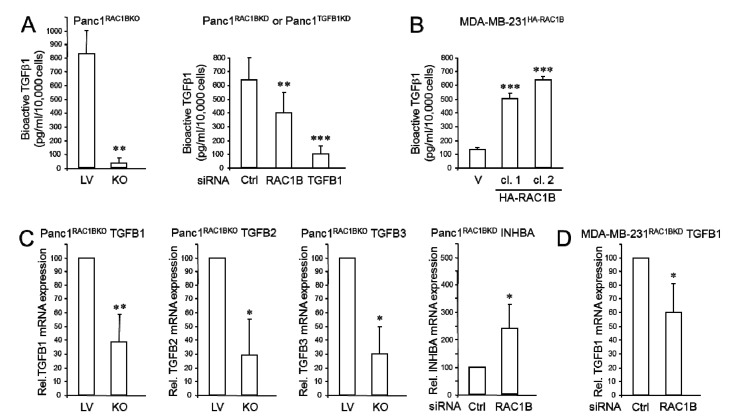
RAC1B depletion controls TGFβ1 gene expression and secretion in Panc1 and MDA-MB-231 cells. (**A**) Concentration of bioactive TGFβ1 in culture supernatants, as measured by ELISA, of Panc1 cells genetically engineered to lack exon 3b of *RAC1* and hence expression of RAC1B (Panc1^RAC1BKO^) or empty lentiviral vector (LV) control cells [7] (827.4 ± 166.9 vs. 35.2 ± 40.7, *n* = 4, *p* = 0.0012, left-hand graph), or Panc^RAC1BKD^ and ctrl cells (393.3 ± 98.2 vs. 264.8.2 ± 98.3, *n* = 3, *p* = 0.0038) or Panc1^TGFB1KD^ and ctrl cells (393.3 ± 98.2 vs. 85.5 ± 38.9, *n* = 3, *p* = 0.029, right-hand graph). Cells were allowed to condition the media for 24 h. (**B**) As in (**A**), except that culture supernatants were retrieved from two individual clones (cl.) of MDA-MB-231 cells stably transfected with HA-RAC1B. Data shown are representative of three assays performed in total (means ± SD from triplicate samples). (**C**) Panc1^RAC1BKO^ cells were subjected to qPCR analyses of *TGFB1* (*n* = 5), *TGFB2* (*n* = 3), or *TGFB3* (*n* = 3), whereas Panc1^RAC1BKD^ cells underwent qPCR analysis of *INHBA*. (**D**) MDA-MB-231^RAC1BKD^ cells were subjected qPCR analysis of *TGFB1*. Data in (**C**,**D**) are the mean ± SD of three different transfections. The asterisks indicate significant differences (* *p* < 0.05; ** *p* < 0.01; *** *p* < 0.001). Successful KO or KD of RAC1B or TGFB1 in Panc1 and MDA-MB-231 cells, and verification of ectopic overexpression of HA-RAC1B in MDA-MB-231 cells is shown in Appendix A.

**Figure 2 cancers-12-03570-f002:**
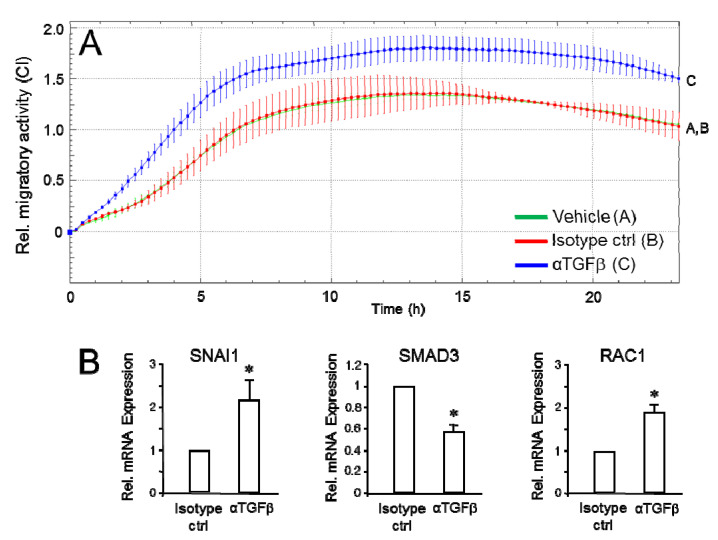
Effects of antibody-mediated neutralization of secreted TGFβ on cell motility. (**A**) MDA-MB-231 cells (10,000/well) were subjected to real-time cell migration assay in culture medium (50 µL) containing only 1% fetal bovine serum (FBS) and in the absence or presence of either vehicle, IgG1 isotype control antibody (Isotype ctrl, 5 µg/mL) or pan-anti-TGFβ1/2/3 antibody (αTGFβ, 5 µg/mL). The assays shown are representative of three assays performed in total. Data represent the mean ± SD of three parallel wells. Differences between IgG1 isotype-treated and αTGFβ-treated cells were first significant at 2:00 and remained so during the course of the assay. The assay shown is representative of three assays. Data represent the mean ± SD of triplicate wells. (**B**) After completion of the assay shown in (**A**), cells were retrieved from the wells and processed for RNA isolation and qPCR analysis of *SNAI1*, *SMAD3* or *RAC1*. *SNAI1* rather than *SNAI2* was chosen because of its closer association with invasive vs. non-invasive breast carcinomas [22] and the stronger efficiency and potency to bind to the *CDH1* promoter and repress its transcription [23]. ECAD expression remained unaltered. Data represent the mean ± SD of triplicate wells. The asterisks indicate statistical significance relative to isotype ctrl treated cells. * *p* < 0.05.

**Figure 3 cancers-12-03570-f003:**
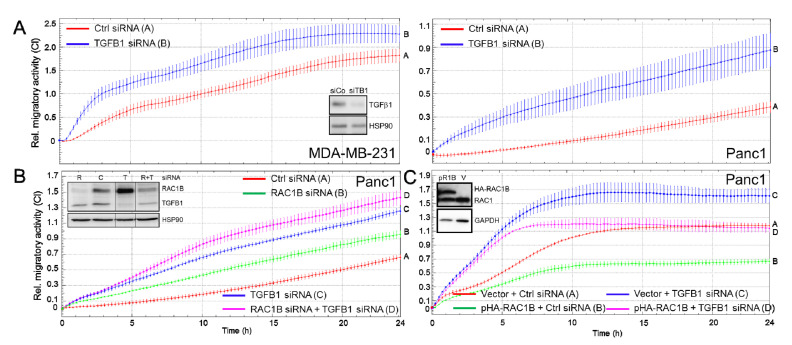
Effect of KD of *TGFB1* on random cell migration in MDA-MB-231 and Panc1 cells. (**A**) MDA-MB-231 (left-hand graph) or Panc1 (right-hand graph) cells were transiently transfected twice with 50 nM each of irrelevant ctrl siRNA or TGFB1 siRNA and 48 h later subjected to real-time cell migration assays on an xCELLigence platform. The data shown are each representative of three assays. (**B**) Panc1 cells were transfected with 50 nM of ctrl siRNA, or 25 nM each of TGFB1 siRNA + ctrl siRNA, RAC1B siRNA + ctrl siRNA, or a combination of TGFB1 and RAC1B siRNA, and evaluated for migratory activity as in (**A**). Differences between curve B and curve C were first significant at 6:00 and all later time points. (**C**) Panc1 cells stably expressing a HA-tagged version of RAC1B (Panc1^HA-RAC1B^) or empty vector (V) control cells were transfected with TGFB1 or ctrl siRNA and subsequently subjected to impedance-based migration assay. Data in (**A**–**C**) are the means ± SD of 3–4 wells and are representative of at least 3 independent experiments. Verification of TGFβ1 KD by ELISA is shown in Figure 1C and elevated *TGFB1* expression in Panc1^HA-RAC1B^ cells in Appendix A. KD of RAC1B or TGFB1, or overexpression of HA-RAC1B, was verified by immunoblotting (insets in panels **A**, **B** and **C**, respectively) and in Appendix A. In the inset in panel B, all signals are from the same blot/exposure from which irrelevant lanes have been removed (indicated by thin vertical lines between lanes 2, 3 and 4, see uncropped blot in Appendix A). C, ctrl; R, RAC1B; T, TGFB1.

**Figure 4 cancers-12-03570-f004:**
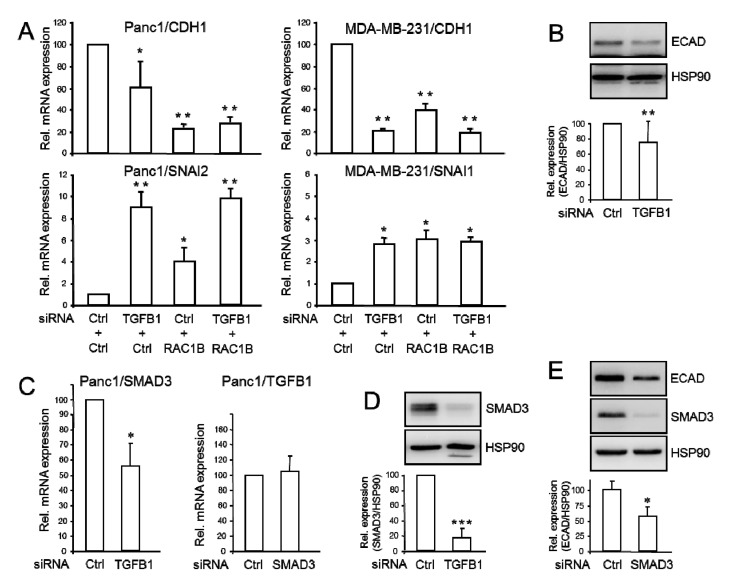
Effects of siRNA-mediated inhibition of TGFβ1 or RAC1B on genes involved in EMT and cell motility. (**A**) Panc1 or MDA-MB-231 cells were transiently transfected twice with 50 nM of either ctrl siRNA, TGFβ1 siRNA, RAC1B siRNA, or a combination of both, as indicated. Forty-eight h later, cells were processed for qPCR analysis of *CDH1* and *SNAI1*, and *GAPDH* as an internal control. (**B**) Panc1^TGFB1KD^ cells were subjected to immunoblot analysis of ECAD, SNAIL, RAC1B, and HSP90 as a loading control. (**C**) As in (**A**), except that Panc1^TGFB1KD^ cells were subjected to amplification of SMAD3 mRNA and Panc1^SMAD3KD^ cells subjected to amplification of TGFβ1 mRNA. (**D**) Immunoblot analysis of SMAD3 in Panc1^TGFB1KD^ cells. (**E**) Immunoblot analysis of ECAD in Panc1^SMAD3KD^ cells. Data in (**A**,**C**) represent the mean ± SD of GOI after normalization with the housekeeping genes (TBP or GAPDH) from three independent transfection experiments. The graphs below the blots in panels (**B**,**D**,**E**) depict data quantification based on densitometric readings from band intensities after normalization with those for HSP90 (mean ± SD; **B**, **D**: *n* = 5, **E**: *n* = 4). The asterisks indicate significant differences (* *p* < 0.05; ** *p* < 0.01; *** *p* < 0.001).

**Figure 5 cancers-12-03570-f005:**
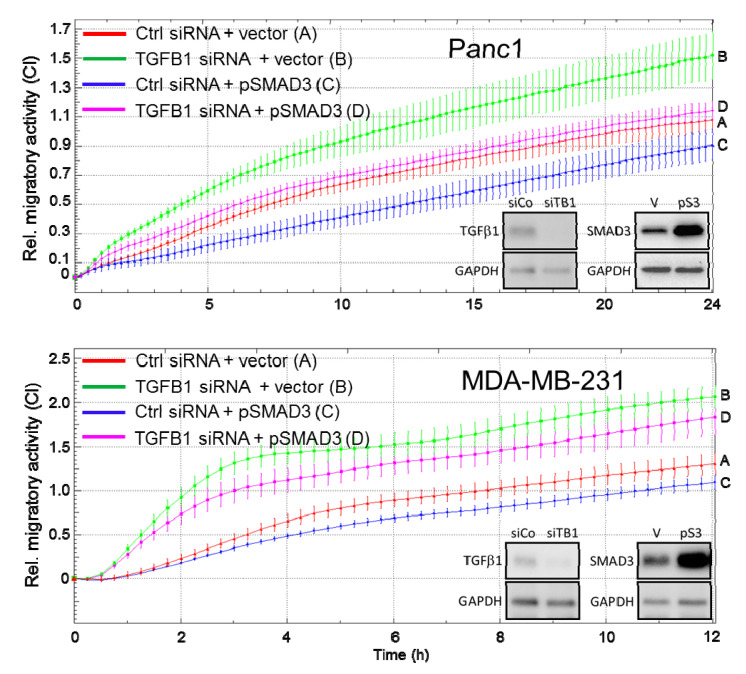
Effect of ectopic SMAD3 expression on cell migration following RNAi-mediated depletion of endogenous TGFβ1. Panc1 or MDA-MB-231 cells were transfected with 50 nM each of either ctrl siRNA or TGFB1 siRNA in combination with either empty pcDNA3.1 vector or pcDNA3.1 containing a SMAD3 cDNA. Forty-eight h after transfection cells were analyzed by real-time cell migration assay. Data are the means ± SD of parallel 3-4 wells and are representative of 3 independent experiments. The differences between curves D and B were first significant (*p* < 0.05) at 3:30 (Panc1) or 3:00 (MDA-MB-231), and for Panc1 cells remained significant over the 24-h assay period. A transient effect of ectopic SMAD3 was also seen in control conditions (curves A vs. curves C, significant between time points 6:30 and 13:00 (Panc1) and 4:00 and 7:30 (MDA-MB-231) in accordance with previous findings [18]). KD of TGFB1 or overexpression of SMAD3 was verified by immunoblotting (insets), siCo, ctrl siRNA; siTB1, TGFB1 siRNA; V, empty vector; pS3, SMAD3 expression vector.

**Figure 6 cancers-12-03570-f006:**
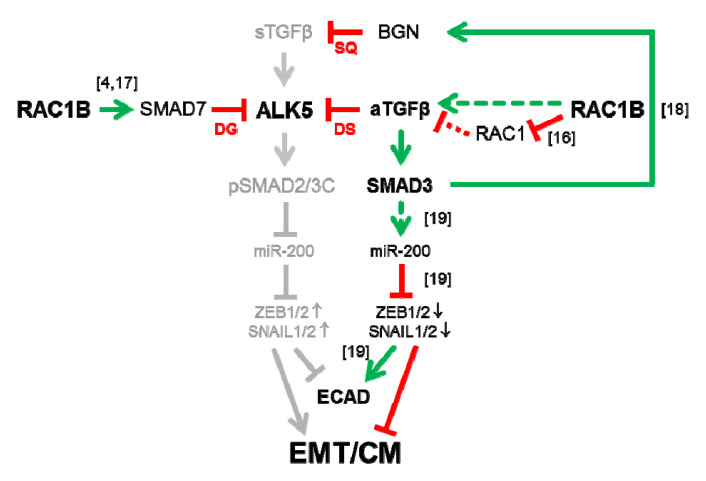
Cartoon illustrating the pathways through which RAC1B targets TGFβ and its receptors for inhibition in PDAC/TNBC-derived cells. Left, RAC1B promotes expression of SMAD7 and SMAD7-mediated degradation (DG) of ALK5. Right, RAC1B—in a RAC1-dependent or independent manner (see Discussion)—promotes expression and secretion of endogenous/autocrine (aTGFβ1). Its occupation of surface receptors induces their rapid internalization and because receptors are replenished only very slowly the cells adopt a refractory state (desensitization, DS) against acute stimulation with paracrine, stromal cell-derived TGFβ (sTGFβ). The sTGFβ activates ALK5 which phosphorylates SMAD2 and SMAD3 at their C-terminus (pSMAD2/3C), while aTGFβ1—in addition to its effects on receptor turnover—upregulates SMAD3 mRNA and protein abundance. Both pathways converge on the miR-200/ZEB1 autoregulatory loop via intermittent transcriptional activation of miR-200 family members to decrease (pSMAD2/3C) or increase (SMAD3) ECAD expression and to stimulate (pSMAD2/3C) or repress (SMAD3) EMT and cell migration (CM) (denoted by arrows or lines). Upper right, The RAC1B-aTGFβ1-SMAD3 pathway may also induce BGN, which prevents receptor activation by sequestration (SQ) of sTGFβ in the TME. Arrows indicate induction/activation and lines repression/inhibition. Stippled red lines denote a still hypothetical interaction. The bracketed numbers next to arrows/lines denote relevant references.

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
