# Peer review of "RAC1B Regulation of TGFB1 Reveals an Unexpected Role of Autocrine TGFβ1 in the Suppression of Cell Motility"

_cancers, 2020, doi:10.3390/cancers12123570_

Round 1
Reviewer 1 Report
This is an interesting paper showing that in PANC1 cells and MDAMB231 cells endogenous or exogenous RAC1B expression suppresses cell migration and elevates TGFb1 production. Knockdown of endogenous TGFb1 increases cell migration and decreases expression of ECAD whilst elevating expression of SNAI1 or SNAI2 according to the cell type. The authors go on to show that autocrine TGFB1 maintains SMAD3 expression levels and that exogenous expression of SMAD3 partially overrides TGFB1 knockdown mediated increase in cell migration. The authors integrate their findings in this paper with their previous findings and those of others into a potential model. On the whole the experiments are well performed and the data is very interesting. I have the following comments that I would like the authors to address.
1) The authors refer to bioactive TGFb1 in their elisa assays. To perform these the supernatants are acidified which can activate TGFb. Is any TGFb1 measurable by bioassay without acid treatment?
2) The authors conclude that because 1D11 treatment inhibits migration but anti-LAP does not that the secreted TGFb must be soluble. Whilst the 1D11 results are good there is no evidence provided to show that the anti-LAP antibody works at all (no positive control). Without this I suggest that the authors should remove this data and any reference to anti-LAP in their manuscript.
3) The authors provide inconsistent verification of knockdown levels or overexpression levels of their knocked down or ectopically expressed proteins at the protein and or QPCR levels in the manuscript. Some figures have this data others do not. For each figure the authors should show knockdown preferably at the protein level or at the QPCR level where this is not possible. It is not possible to judge the level of ectopic expression compared to endogenous levels of SMAD3 or RAC1b.
4) Given the author's findings on differentiation status it would be good to assess the morphology of the cells following knockdown and ectopic expression.
5) It would be good to assess the effects of knockdown and ectopic expression on the proliferative capacity of the cells in the same 1% serum conditions used for the migration experiments
6) The endogenous autocrine TGFb findings are exciting. Do the authors think these are independent of ALK5 given their model in Figure 6? What happens to the cells if you knockdown ALK5 or treat the cells with an ALK5 kinase inhibitor?
7) The authors should indicate which ALK5 antibody they used in the materials and methods section
Author Response
Dear Editor, dear Yolanda:
This letter of submission is accompanied by our revised manuscript entitled:
“RAC1B regulation of TGFB1 reveals an unexpected role of autocrine TGFβ1 in the suppression of cell motility”
We are indebted to the reviewers for their valuable comments and suggestions and have done our best to incorporate these into the revised version of our manuscript (highlighted in the “track changes” mode). We believe that the reviewers’ critiques have substantially improved the quality of our manuscript and we are looking forward to its final acceptance in Cancers.
Sinerely yours,
Hendrik Ungefroren
This is an interesting paper showing that in PANC1 cells and MDAMB231 cells endogenous or exogenous RAC1B expression suppresses cell migration and elevates TGFb1 production. Knockdown of endogenous TGFb1 increases cell migration and decreases expression of ECAD whilst elevating expression of SNAI1 or SNAI2 according to the cell type. The authors go on to show that autocrine TGFB1 maintains SMAD3 expression levels and that exogenous expression of SMAD3 partially overrides TGFB1 knockdown mediated increase in cell migration. The authors integrate their findings in this paper with their previous findings and those of others into a potential model. On the whole the experiments are well performed and the data is very interesting. I have the following comments that I would like the authors to address.
1) The authors refer to bioactive TGFb1 in their elisa assays. To perform these the supernatants are acidified which can activate TGFb. Is any TGFb1 measurable by bioassay without acid treatment?
Response: Yes! In supernatants from parental Panc1 cells conditioned for 48 h the amount of bioactive TGFb1 without acid treatment is approx. 1-5% of the amount of acidified supernatant.
2) The authors conclude that because 1D11 treatment inhibits migration but anti-LAP does not that the secreted TGFb must be soluble. Whilst the 1D11 results are good there is no evidence provided to show that the anti-LAP antibody works at all (no positive control). Without this I suggest that the authors should remove this data and any reference to anti-LAP in their manuscript.
Response: We totally agree with this point. Therefore, and as suggested, all data with the anti-LAP antibody have been removed from the main text and the Supplementary material.
3) The authors provide inconsistent verification of knockdown levels or overexpression levels of their knocked down or ectopically expressed proteins at the protein and or QPCR levels in the manuscript. Some figures have this data others do not. For each figure the authors should show knockdown preferably at the protein level or at the QPCR level where this is not possible. It is not possible to judge the level of ectopic expression compared to endogenous levels of SMAD3 or RAC1b.
Response: As requested, verification of knockdown and overexpression levels has been added for both cell lines and to all figures.
4) Given the author's findings on differentiation status it would be good to assess the morphology of the cells following knockdown and ectopic expression.
Response: The morphology of Panc1-RAC1B-KD cells has been shown in another publication (Ref. 16, Fig. 4A). Without exogenous TGFb1 treatment, differences in morphology between RAC1B KD and control cells were small and differences were not significant (percentage of spindle-shaped cells: 17.5±8 vs. 10.1±4). However, the situation was different after a 48-h treatment with 5 ng/ml recombinant (rec.) human TGFb1, when the number of elongated cells increased to a much greater extent in the RAC1B-KD cells than in the control cells (Ref. 16, Fig. 4A). Likewise, the number of spindle-shaped cells in the non-rec. TGFb1-treated TGFB1-KD population did not differ from the control population, but did so after stimulation for 48 h with 5 ng/ml rec. TGFb1. We plan to include these data in separate manuscript to be submitted to “Cancers” soon.
5) It would be good to assess the effects of knockdown and ectopic expression on the proliferative capacity of the cells in the same 1% serum conditions used for the migration experiments
Response: As requested, we have performed cell counting experiments in Panc1 cells following transfection with Panc1-RAC1B-KD or TGFB1-KD cells, alone and in combination. These revealed that with both siRNAs the number of cells dropped significantly to roughly the same extent when compared to irrelevant control siRNA transfectants. Moreover, cell numbers for the combined knockdown were not different from those of the single knockdowns. With respect to RAC1B, this is in line with what we observed previously for Panc1-RAC1B knockout cells (Ref. 4). These data further support the idea that both RAC1B and autocrine TGFb1 mediate proliferative effects with autocrine TGFb1 acting downstream of RAC1B. The pro-proliferative effect of autocrine TGFb has been demonstrated previously in MDA-MB-231 cells (Ref. 7). We have displayed these data with Panc1 cells in the new Supplementary Figure S3.
6) The endogenous autocrine TGFb findings are exciting. Do the authors think these are independent of ALK5 given their model in Figure 6? What happens to the cells if you knockdown ALK5 or treat the cells with an ALK5 kinase inhibitor?
Response: This is a very interesting issue. To study a possible involvement of ALK5, we have performed migration assays with Panc1-TGFB1-KD cells in the presence or absence of SB431542 (1 µM). Results show that treatment of TGFB1-KD cells with SB431542 did not significantly change migratory activities, however, there was a trend towards an increase. Based on these data, we conclude that the inhibitory effect of autocrine TGFb1 on cell migration is indeed independent of ALK5. We have briefly mentioned this finding in the Results section (end of first paragraph in section 2.3).
7) The authors should indicate which ALK5 antibody they used in the materials and methods section
Response: As requested, product data for the anti-ALK5 and anti-TGFb1 antibodies used for the immunoblots have been added to the Material & Methods, section 4.2.
Reviewer 2 Report
This is a well written interesting study with data supporting a potential tumor suppressive effect for RAC1b, that acting by increasing the autocrine secretion of TGFβ1 has an inhibitory impact on cell migration. Still, there are several points that the authors should clarify.
General comments:
In most experiments, the levels of KO, KD or overexpressed proteins are not displayed. Showing the up- or down-regulated protein levels is quite relevant to allow monitoring and comparing the effects observed under different conditions in different cellular models.
It is also quite relevant to address whether the KO of RAC1b impact on the levels of the active pool of endogenous RAC1, since an increase on RAC1 activity can produce effects similar to that attributed to the absence of RAC1b.
In addition, it is also important to monitor the levels of the RAC1b KD upon siRNA in comparison to endogenous RAC1 levels (total RAC1 pool).
Specific comments:
Figure 1:
The levels of KO RAC1b, KD RAC1b and TGFβ1 as well as of ectopic overexpression of HA-RAC1b should be displayed.
The reason for testing INHBA on Panc1 RAC1bKD is unclear. This should be explained in the text for the sake of clarity.
Figure 2 and Supplementary Figure S2:
The reason for testing SNAI1 in MDA-MB-231 but in Panc1 the SNAI2 is unclear. This should be explained in the text for the sake of clarity. The authors should also explain the reason why RAC1 and ECAD were not tested on both MDA-MB-231 and Panc1 cell lines.
The impact of antibody-mediated neutralization of secreted and membrane-bound autocrine TGFβ on RAC1b endogenous expression should be also addressed.
Figure 3:
A Western blot (Wb) showing the levels of TGFB1 KD and RAC1B + TGFB1 double KD upon siRNA should be presented. Also, endogenous RAC1 levels should be shown.
Figure legend should be revised: no ectopic overexpression of TGFB1 was addressed in this experiment; Wb panels were not described.
Figure 4:
Again, although the monitoring of TGFB1 KD by Wb is represented in Supplementary Figure S4, this is not the case for RAC1b KD and should also be shown.
Again, the authors should explain why choosing to analyze the expression of the SNAI1 gene in MDA-MB-231 and of SNAI2 in Panc 1 cells.
Figure 5:
A Wb monitoring SMAD 3 ectopic expression is not represented and should be shown.
The monitoring of TGFB1 KD by Wb should be shown also.
The effect of ectopic SMAD3 expression on cell migration following RNAi-mediated depletion of endogenous TGFβ1 (comparing B and D curves) seem to be of the same order of magnitude as that observed in control conditions (comparing A and C curves). Statistical significance for the difference between curves A and C must be presented. Nonetheless, this experiment does not unequivocally show that ectopic SMA3 expression rescues the cells from a TGFB1 KD-induced increase in migratory activity. Rather, SMAD3 can have an antimigratory effect independent of TGFβ1 downstream signaling. To show that SMA3 acts downstream from TGFβ1, the opposite experience should be carried out: suppression of cell migration should be induced by TGFβ1 in the presence od endogenous SMA3 and upon SMA3 depletion. The absence of SMA3 should impair the TGFβ1- induced decrease in migratory activity.
Discussion
Despite the authors having already collected solid evidence supporting a tumor suppressive role for RAC1b in some malignant contexts, the tumor-promoting role of this GTPase is also strongly documented in the literature. These two seemingly opposing facets of RAC1b should be discussed and conciliated in the discussion section.
Author Response
Dear Editor, dear Yolanda:
This letter of submission is accompanied by our revised manuscript entitled:
“RAC1B regulation of TGFB1 reveals an unexpected role of autocrine TGFβ1 in the suppression of cell motility”
We are indebted to the reviewers for their valuable comments and suggestions and have done our best to incorporate these into the revised version of our manuscript (highlighted in the “track changes” mode). We believe that the reviewers’ critiques have substantially improved the quality of our manuscript and we are looking forward to its final acceptance in Cancers.
Sinerely yours,
Hendrik Ungefroren
This is a well written interesting study with data supporting a potential tumor suppressive effect for RAC1b, that acting by increasing the autocrine secretion of TGFβ1 has an inhibitory impact on cell migration. Still, there are several points that the authors should clarify.
General comments:
1) In most experiments, the levels of KO, KD or overexpressed proteins are not displayed. Showing the up- or down-regulated protein levels is quite relevant to allow monitoring and comparing the effects observed under different conditions in different cellular models.
Response: This request was also raised by reviewer 1. As requested, we have now added the immunoblot images with protein levels in KO and KD cells for all experiments to either the main figures or the Supplementary figures.
2) It is also quite relevant to address whether the KO of RAC1b impact on the levels of the active pool of endogenous RAC1, since an increase on RAC1 activity can produce effects similar to that attributed to the absence of RAC1b.
Response: This is a very good suggestion. Although we observed that HA-RAC1B ectopic expression in MDA-MB-231 cells on its own increased autocrine TGFb1 secretion (see Fig. 1B), we found previously in Panc1 cells that RAC1B-KD caused a small increase in total levels of RAC1 (Ref. 16: immunoblot in Figure 4B). We agree with this reviewer that it is possible that this increase in RAC1 protein can mediate the effect of RAC1B-KD on autocrine TGFb1 production/secretion and cell migration. However, in order to fully analyze this issue, many more experiments have to be performed besides measuring the amount of active RAC1 in the total pool of endogenous RAC1, i.e. rescue experiments with selective KD of RAC1. Therefore, we believe that rigorously elucidating the role of RAC1 in mediating the effect of RAC1B on autocrine TGFb1 production is beyond the scope of this manuscript.
However, to elucidate if RAC1 accounts for the effect of RAC1B on SMAD3 expression (the executor of the migration-inhibiting effect), we transiently and ectopically expressed in Panc1 cells RAC1-Q61L, a constitutively active RAC1 mutant. This mutant although highly active in phosphorylating p38 MAPK, failed to suppress SMAD3 abundance as measured by immunoblotting. This suggests that activated RAC1 alone is unable to mimic the effect of RAC1B on SMAD3 and probably also on autocrine TGFb1.
We have briefly discussed the possible involvement of RAC1 in RAC1B function in the Discussion section (3rd paragraph) and have modified Figure 6 accordingly.
3) In addition, it is also important to monitor the levels of the RAC1b KD upon siRNA in comparison to endogenous RAC1 levels (total RAC1 pool).
Response: Very good suggestion! Already done in a previous study (see Ref. 16: Figure 4B).
Specific comments:
1) Figure 1: The levels of KO RAC1b, KD RAC1b and TGFβ1 as well as of ectopic overexpression of HA-RAC1b should be displayed.
Response: As requested, these protein levels are shown now in Supplementary Figure S1, panel B.
2) The reason for testing INHBA on Panc1 RAC1bKD is unclear. This should be explained in the text for the sake of clarity.
Response: As requested, a short explanatory sentence has been added (2nd paragraph in section 2.1).
3) Figure 2 and Supplementary Figure S2: The reason for testing SNAI1 in MDA-MB-231 but in Panc1 the SNAI2 is unclear. This should be explained in the text for the sake of clarity. The authors should also explain the reason why RAC1 and ECAD were not tested on both MDA-MB-231 and Panc1 cell lines.
Response: As requested, explanations have been added to the legend of Figure 2 (for SNAI1 and ECAD) and the legend to Supplementary Figure S2 (for SNAI2 and RAC1).
4) The impact of antibody-mediated neutralization of secreted and membrane-bound autocrine TGFβ on RAC1b endogenous expression should be also addressed.
Response: As requested, we have assessed the effect of anti-TGFβ1/2/3 antibody on RAC1B at the mRNA level by qPCR but failed to detect an effect. This was not surprising, since autocrine TGFb1 is located downstream of RAC1B in the proposed pathway. All data with anti-LAP (recognizing membrane-bound TGFb) have been removed in response to a request from reviewer 1.
5) Figure 3: A Western blot (Wb) showing the levels of TGFB1 KD and RAC1B + TGFB1 double KD upon siRNA should be presented. Also, endogenous RAC1 levels should be shown.
Response: As requested, we have included immunoblot images with the levels of TGFB1, RAC1B in single and combined KD as insets in Figure 3B. With respect to RAC1, we prefer not to show endogenous RAC1 levels as outlined above for the reasons stated in points 2 and 3. This will be the subject of a separate publication.
6) Figure legend should be revised: no ectopic overexpression of TGFB1 was addressed in this experiment; Wb panels were not described.
Response: As requested, the figure legend has been corrected. The WB insets are mentioned at the end of the figure legend.
7) Figure 4: Again, although the monitoring of TGFB1 KD by Wb is represented in Supplementary Figure S4, this is not the case for RAC1b KD and should also be shown.
Response: This has been rectified as requested and data are included as an inset in Figure 3B.
8) Again, the authors should explain why choosing to analyze the expression of the SNAI1 gene in MDA-MB-231 and of SNAI2 in Panc 1 cells.
Response: This has already been done above (see point 3).
9) Figure 5: A Wb monitoring SMAD 3 ectopic expression is not represented and should be shown. The monitoring of TGFB1 KD by Wb should be shown also.
Response: As requested, immunoblots of SMAD3 and TGFB1 from the cells subjected to migration assays have been added as insets to Figure 5.
10) The effect of ectopic SMAD3 expression on cell migration following RNAi-mediated depletion of endogenous TGFβ1 (comparing B and D curves) seem to be of the same order of magnitude as that observed in control conditions (comparing A and C curves). Statistical significance for the difference between curves A and C must be presented.
Response: Statistical evaluation of the differences between curves A and C has been performed. Differences were found to be significant during a subperiod of the assays with both cell lines as reported previously for Panc1 cells (Ref. 18). This has been indicated in the legend to Figure 5.
Nonetheless, this experiment does not unequivocally show that ectopic SMA3 expression rescues the cells from a TGFB1 KD-induced increase in migratory activity. Rather, SMAD3 can have an antimigratory effect independent of TGFβ1 downstream signaling. To show that SMA3 acts downstream from TGFβ1, the opposite experience should be carried out: suppression of cell migration should be induced by TGFβ1 in the presence of endogenous SMA3 and upon SMA3 depletion. The absence of SMA3 should impair the TGFβ1- induced decrease in migratory activity.
Response: It is true that SMAD3 has an antimigratory effect independent of TGFβ1 downstream signaling (see Ref. 18). However, this effect is weaker than SMAD3’s effect on TGFB1-KD cells (see legend to Figure 5). Unfortunately, the reviewers’ suggestion is unlikely to work since rec. TGFβ1 stimulates rather than inhibits migration in both Panc1 and MDA-MB-231 cells (Refs. 4, 6, 16, 18, 19, 24)! In fact, TGFB1-KD enhances the stimulatory effect of rec. TGFβ1 on cell migration. This is part of a separate manuscript to be submitted soon.
11) Discussion: Despite the authors having already collected solid evidence supporting a tumor suppressive role for RAC1b in some malignant contexts, the tumor-promoting role of this GTPase is also strongly documented in the literature. These two seemingly opposing facets of RAC1b should be discussed and conciliated in the discussion section.
Response: As requested, we have included a short paragraph on to the tumor-promoting role of RAC1B and the tumor-suppressive role for RAC1B in the context of pro-invasive TGFβ1 signaling. However, we have moved this part to the 2nd paragraph of the Introduction section, given the already great length of the Discussion section.
Reviewer 3 Report
TGF-b1 is a potent inducer of Epithelial-Mesenchymal Transition, which has been implicated in the progression of epithelial based cancers. In the current work, the authors provide evidence for a paradigm-shifting perspective on TGF-β1’s role in EMT and as a suppressor of cell motility. The authors present convincing evidence that knocking down RAC1B inhibits TGFB1, TGFB2, and TGFB3 expression. They next convincingly demonstrate that autocrine TGF-β1 plays an anti-migratory protective function in both MDA-MBA-231 and Panc1 cells. They further characterize this by assessing the effects of knockdown of autocrine TGF-beta1 on other EMT markers (CDH1 (E-Cadherin), SNAI1/2 and SMAD3) and demonstrate a shift towards a mesenchymal phenotype in response to TGF-beta1 inhibition. Finally, they assessed the effects of SMAD3 knockdown and overexpression and establish that SMAD3 (induced downstream of TGF-β1 signaling) can counteract the pro-migratory effects of TGFB1 knockdown. This culminates into a proposed regulation scheme separating the effects of autocrine and stromal TGF-β1 to explain the differences from canonical TGF-β1-regulated EMT.
My biggest concern with this manuscript is the way in which the TGF-b1 complex is discussed. TGF-b1 is secreted as a soluble complex that contains LTBP-1, LAP, and TGF-b1. This complex may bind to fibrils in the extracellular matrix, but is also found in soluble form within conditioned culture media. The paper makes several statements regarding the TGF-beta complex that are either misleading or incorrect. First, when using LAP neutralizing antibodies, the authors comment that these antibodies will specifically target membrane-bound TGF-beta1. I presume that the authors meant “matrix-bound” instead of “membrane bound” --this should be corrected. But even with this correction, the statement is not accurate. Soluble TGF-beta is bound to both LAP and LTBP-1, and so using a LAP antibody will not specifically target only matrix-bound TGF-beta. More than likely, the LAP antibody is not blocking the active site of TGF-beta, and is thus more of a negative control than anything else.
Similarly, the authors distinguish in multiple cases between stromal TGF-beta and autocrine TGF-beta, but there is no rationale for there being differences between these two nor differences between how the cell may respond to these. It is possible that the authors are suggesting a difference between soluble TGF-beta and matrix-bound TGF-beta; this terminology would be more appropriate than stromal versus autocrine, as TGF-beta, regardless of cell source, is secreted in a complex with LTBP-1 and LAP and may bind to the matrix.
Additionally, the following concerns are also noted:
In Figure 4, Snai2 expression is shown for Panc1 cells, but Snai1 is shown for MDA-MB-231 cells. Please explain this inconsistency.
In Figure 1, bioactive TGF-β1 in cell culture supernatant may not be completely reflective of all secreted TGF-β1 as it may bind to the extracellular matrix if sufficient fibronectin is assembled.
Figure 3 is titled as “Effect of KD or ectopic overexpression of TGFB1…” TGFB1 is knocked down, but it not directly overexpressed. This should be corrected.
Author Response
Dear Editor, dear Yolanda:
This letter of submission is accompanied by our revised manuscript entitled:
“RAC1B regulation of TGFB1 reveals an unexpected role of autocrine TGFβ1 in the suppression of cell motility”
We are indebted to the reviewers for their valuable comments and suggestions and have done our best to incorporate these into the revised version of our manuscript (highlighted in the “track changes” mode). We believe that the reviewers’ critiques have substantially improved the quality of our manuscript and we are looking forward to its final acceptance in Cancers.
Sinerely yours,
Hendrik Ungefroren
TGF-b1 is a potent inducer of Epithelial-Mesenchymal Transition, which has been implicated in the progression of epithelial based cancers. In the current work, the authors provide evidence for a paradigm-shifting perspective on TGF-β1’s role in EMT and as a suppressor of cell motility. The authors present convincing evidence that knocking down RAC1B inhibits TGFB1, TGFB2, and TGFB3 expression. They next convincingly demonstrate that autocrine TGF-β1 plays an anti-migratory protective function in both MDA-MBA-231 and Panc1 cells. They further characterize this by assessing the effects of knockdown of autocrine TGF-beta1 on other EMT markers (CDH1 (E-Cadherin), SNAI1/2 and SMAD3) and demonstrate a shift towards a mesenchymal phenotype in response to TGF-beta1 inhibition. Finally, they assessed the effects of SMAD3 knockdown and overexpression and establish that SMAD3 (induced downstream of TGF-β1 signaling) can counteract the pro-migratory effects of TGFB1 knockdown. This culminates into a proposed regulation scheme separating the effects of autocrine and stromal TGF-β1 to explain the differences from canonical TGF-β1-regulated EMT.
1) My biggest concern with this manuscript is the way in which the TGF-b1 complex is discussed. TGF-b1 is secreted as a soluble complex that contains LTBP-1, LAP, and TGF-b1. This complex may bind to fibrils in the extracellular matrix, but is also found in soluble form within conditioned culture media. The paper makes several statements regarding the TGF-beta complex that are either misleading or incorrect. First, when using LAP neutralizing antibodies, the authors comment that these antibodies will specifically target membrane-bound TGF-beta1. I presume that the authors meant “matrix-bound” instead of “membrane bound” --this should be corrected. But even with this correction, the statement is not accurate. Soluble TGF-beta is bound to both LAP and LTBP-1, and so using a LAP antibody will not specifically target only matrix-bound TGF-beta. More than likely, the LAP antibody is not blocking the active site of TGF-beta, and is thus more of a negative control than anything else.
Response: We thank the reviewer for these pieces of information and sincerely apologize that our statements were misleading or incorrect. In response to a request from reviewer 1, we have removed all data with the LAP antibody as well as the statements regarding the TGFb complex.
2) Similarly, the authors distinguish in multiple cases between stromal TGF-beta and autocrine TGF-beta, but there is no rationale for there being differences between these two nor differences between how the cell may respond to these. It is possible that the authors are suggesting a difference between soluble TGF-beta and matrix-bound TGF-beta; this terminology would be more appropriate than stromal versus autocrine, as TGF-beta, regardless of cell source, is secreted in a complex with LTBP-1 and LAP and may bind to the matrix.
Response: This is correct. We have therefore replaced the term “stromal TGFβ” by “stromal cell-derived TGFβ” or “paracrine TGFβ”. However, we found that rec. TGFβ (assumed to be the in vitro substitute for the stromal cell-derived TGFβ in vivo) has the opposite effect on cell migration than autocrine TGFβ. This is to be published soon in a separate manuscript.
Additionally, the following concerns are also noted:
3) In Figure 4, Snai2 expression is shown for Panc1 cells, but Snai1 is shown for MDA-MB-231 cells. Please explain this inconsistency.
Response: This issue was also raised by Reviewer 2. As requested, explanations have been added to the legend of Figure 2 (for SNAI1 and ECAD) and the legend to Supplementary Figure S2 (for SNAI2 and RAC1).
4) In Figure 1, bioactive TGF-β1 in cell culture supernatant may not be completely reflective of all secreted TGF-β1 as it may bind to the extracellular matrix if sufficient fibronectin is assembled.
Response: We thank the reviewer for this piece of information. We have briefly mentioned this in the first paragraph of the Results section.
5) Figure 3 is titled as “Effect of KD or ectopic overexpression of TGFB1…” TGFB1 is knocked down, but it not directly overexpressed. This should be corrected.
Response: As requested, this error has been corrected.
Reviewer 4 Report
Excellent manuscript, skillfully written. I have two minor comments.
Line 624-626
However, during the course of this study we realized that in mesenchymal-type cancer cells from pancreas and breast RAC1B-induced autocrine TGFβ can block ECAD expression and cell motility through a pathway that involves SMAD3 and presumably SMAD3-dependent induction of miR-200.
My understanding is that Rac1B induced autocrine TGFbeta results in increased (not block) ECAD expression (Figure 4B and diagramm). Please make sure line 624-625 statement is correct.
Fig1. Graph title Panc1RAC1BKD+Panc1TGFB1KD suggests double knockdown of Rac1 and TBFB1 in the same cell line. Alternative title - "Panc1RAC1BKD or Panc1TGFB1KD"
Author Response
Dear Editor, dear Yolanda:
This letter of submission is accompanied by our revised manuscript entitled:
“RAC1B regulation of TGFB1 reveals an unexpected role of autocrine TGFβ1 in the suppression of cell motility”
We are indebted to the reviewers for their valuable comments and suggestions and have done our best to incorporate these into the revised version of our manuscript (highlighted in the “track changes” mode). We believe that the reviewers’ critiques have substantially improved the quality of our manuscript and we are looking forward to its final acceptance in Cancers.
Sinerely yours,
Hendrik Ungefroren
Excellent manuscript, skillfully written. I have two minor comments.
1) Line 624-626
However, during the course of this study we realized that in mesenchymal-type cancer cells from pancreas and breast RAC1B-induced autocrine TGFβ can block ECAD expression and cell motility through a pathway that involves SMAD3 and presumably SMAD3-dependent induction of miR-200. My understanding is that Rac1B induced autocrine TGFbeta results in increased (not block) ECAD expression (Figure 4B and diagramm). Please make sure line 624-625 statement is correct.
Response: This is certainly true and we have corrected this statement.
2) Fig1. Graph title Panc1RAC1BKD+Panc1TGFB1KD suggests double knockdown of Rac1 and TBFB1 in the same cell line. Alternative title - "Panc1RAC1BKD or Panc1TGFB1KD"
Response: This is correct and we have changed the title according to the reviewer’s suggestion.
Round 2
Reviewer 1 Report
I thank the authors for addressing my comments which I believe has improved the study. I congratulate them on an interesting paper.
Author Response
We sincerely thank this reviewer for his enthusiastic comments and his congratulations.
Reviewer 2 Report
The revised article in this manuscript has been significantly improved. However, some of the responses provided did not fully address the concerns raised by the reviewer and there are still some issues that need to be clarified. Particularly:
Author´s answer to point 2- The small increase in total levels of RAC1 previously (REF16) found to be caused RAC1B-KD by in Panc1 and the possibility that this increase in RAC1 protein can mediate the effect of RAC1B-KD on autocrine TGFb1 production/secretion and cell migration should be mentioned and discussed in the present manuscript.
Furthermore, this reviewer was unable to locate the mentioned data regarding the experiment showing the impact of RAC1-Q61L in SMAD3 expression.
Author´s answer to point 3- Data from the previous study concerning the monitoring of RAC1b KD levels in comparison to endogenous RAC1 levels (total RAC1 pool) should be mentioned in the present manuscript.
Author´s answer to specific comment 1-
Captions of Supplementary Figure S1, panel B, third blott: where shows RAC1b must show TGFB1 instead.
Author´s answer to specific comment 5-
Figure 3B: The image displayed representing the western blot (WB) is of poor quality. The control and the target are in different wb / different exposures, not allowing a reliable evaluation of the variations. The lane designated as M is likely not to correspond to the loading of the molecular weight marker, as indicated in the figure legend.
Author Response
We are delighted that this reviewer considered our manuscript to be much improved and we very much appreciate his additional comments that we have addressed as follows:
The revised article in this manuscript has been significantly improved. However, some of the responses provided did not fully address the concerns raised by the reviewer and there are still some issues that need to be clarified. Particularly:
1) Author´s answer to point 2- The small increase in total levels of RAC1 previously (REF16) found to be caused RAC1B-KD by in Panc1 and the possibility that this increase in RAC1 protein can mediate the effect of RAC1B-KD on autocrine TGFb1 production/secretion and cell migration should be mentioned and discussed in the present manuscript.
Response: As mentioned in our answer to point 2, we have briefly mentioned this observation and have discussed the possible involvement of RAC1 in RAC1B function in the Discussion section (3rd paragraph of the Discussion section in the revised version). We have also modified the cartoon in Figure 6, top right, to indicate that inhibition of RAC1 by RAC1B (shown in Ref. 16) could relieve a possible suppressive effect of RAC1 on autocrine TGF-beta1 production/secretion, ultimately resulting in upregulation of autocrine TGF-beta1 by RAC1B.
2) Furthermore, this reviewer was unable to locate the mentioned data regarding the experiment showing the impact of RAC1-Q61L in SMAD3 expression.
Response: We apologize for not having supplied these data in the revised manuscript. The SMAD3 blot of Panc1 cells with ectopic expression of RAC1-Q61L is now shown in the Supplementary material as Figure S6 and these data are also discussed in the third paragraph of the Discussion section.
3) Author´s answer to point 3- Data from the previous study concerning the monitoring of RAC1b KD levels in comparison to endogenous RAC1 levels (total RAC1 pool) should be mentioned in the present manuscript.
Response: Please, see above (1.) since this issue is related to our answer to point 2. The previous study (Ref. 16) has been referenced there (3rd paragraph of the Discussion section).
4) Author´s answer to specific comment 1 - Captions of Supplementary Figure S1, panel B, third blot: where shows RAC1b must show TGFB1 instead.
Response: That is true and we are sorry for this confusion. The blot is correct, but the labeling was wrong. We have corrected this error.
5) Author´s answer to specific comment 5 - Figure 3B: The image displayed representing the western blot (WB) is of poor quality. The control and the target are in different wb / different exposures, not allowing a reliable evaluation of the variations. The lane designated as M is likely not to correspond to the loading of the molecular weight marker, as indicated in the figure legend.
Response: We apologize that we missed to indicate that this image represents the same blot, from which irrelevant lanes have been removed. Indeed, signals in the inset of panel 3B are all from the same blot/exposure, but irrelevant lanes have been cropped between lanes 2 and 3, and lanes 3 and 4 (as indicated by thin vertical lines). To demonstrate this, we have supplied the uncropped blot, including the lane with the molecular weight marker, in Supplementary material as Figure S7. For this reason, we have removed the lane with the MW from the blot in Figure 3B for the sake of clarity and to reduce complexity of this figure.
Reviewer 3 Report
All concerns from the original submission have been adequately addressed.